# LABEL REFINING: A SEMI-SUPERVISED METHOD TO EXTRACT VOICE CHARACTERISTICS WITHOUT GROUND TRUTH

## ABSTRACT

A characteristic is a distinctive trait shared by a group of observations which may be used to identify them. In the context of voice casting for audiovisual productions, characteristic extraction has an important role since it can help explaining the decisions of a voice recommendation system, or give modalities to the user with the aim to express a voice search request. Unfortunately, the lack of standard taxonomy to describe comedian voices prevents the implementation of an annotation protocol. To address this problem, we propose a new semi-supervised learning method entitled Label Refining that consists in extracting refined labels (e.g. vocal characteristics) from known initial labels (e.g. character played in a recording). Our proposed method first suggests using a representation extractor based on the initial labels, then computing refined labels using a clustering algorithm to finally train a refined representation extractor. The method is validated by applying Label Refining on recordings from the MassEffect 3 video game. Experiments show that, using a subsidiary corpus, it is possible to bring out interesting voice characteristics without any a priori knowledge.

## 1 INTRODUCTION

Localization of a multimedia work, such as a video game or a motion picture, consists in making changes and adaptations to the original work so that international distributors can market the resulting product in a target country. For this purpose, companies need to translate their works into a target language. Various options are available to them to give substance to their translation. Subtitling is the simplest and cheapest option, but not the most practical one for spectators: a large audience is more comfortable with hearing speech, in general in their mother tongue, rather than reading subtitles while hearing speech in another language. Dubbing, which is more expensive and takes longer to set up, better retains the audience's immersion by replacing the original voices with dubbing comedian voices. It involves a voice selection process, which means choosing among several candidate voices in a target language. This selection is referred to as voice casting.

With the emergence of streaming platforms (e.g. Netflix, Disney+, Amazon Prime Video) and the rapid development of the video game industry, the number of works to be localized is increasing dramatically. As a result, more and more actors are available on the voice market. Vocal Casting is an artisanal process that requires a lot of precision and is carried out by human operators. As it is, it cannot be applied to a very large number of actors. Operators very often rely on their memory, which is limited to a small number of actors with whom they have worked and in whom they have confidence. However, this operation can degrade the immersion of the audience since the most oft-used voices become easily recognizable. They then become references and create cognitive biases in the public's perception of the character. Automatic actor search and recommendation tools, based on voice processing, would help operators find new actors who would then enrich the vocal diversity of the works to the delight of the audience.

In this article, we wish to address some of the complexity of voice casting and we start with the voices of professional actors playing characters for the gaming industry. The articles Gresse et al. (2020; 2019; 2017); Malik & Nguyen (2021); Obin & Roebel (2016); Obin et al. (2014); Quillot et al. (2021b;a) cited in this paper are the only ones dealing with the speech-processing-based automation

of Voice Casting in the literature to our knowledge. We rely on Gresse et al. (2020), a recent work which defines the $p$-vector, a neural-network-based representation of the voice dedicated to the character aspects in acted voices. The application context in Gresse et al. (2020) (like in related previous works Gresse et al. (2017; 2019)) is voice casting for audio dubbing. The final objective of this work is to assess how close an actor – acting in a target language – is to the voice of a given character, speaking in a source language. This can be referred to as character-based voice similarity. In contrast to Obin et al. (2014); Obin & Roebel (2016), which proposed a voice similarity system for the acted voice based on data labeled with speech classes (age, gender, emotions ...) by an expert, the $p$-vector approach does not use any expert labels. This representation is learnt from audio associated with the played character label, without any additional information. Also, the authors of Quillot et al. (2021a) recently posed the following hypothesis: when a character is played, the actor adds to the voice some information that characterizes the character and is independent of the speaker identity. They then introduced SICI (Speaker-Independent Character Information) and showed that there is speaker-independent character information in the acted voice and that it is present in $p$-vectors pre-trained on a speaker task Quillot et al. (2021b).

Although these approaches deal with final works (video games), another one consists in working directly on the decision data from Artistic Directors. Recently, researchers from Warner Bros. evaluated their models on this kind of data. Unfortunately, these data are sensitive and their acquisition is not trivial since it requires to work on the critical voice casting process. This is why we decided to position ourselves in a task quite different from Malik & Nguyen (2021), since we do not use Artistic Director decisions.

Voice similarity systems alone are not enough for Voice Casting operators. They do not trust the computed recommendations and need a more complete interface to interact with the system. Many works propose explainability approaches Došilović et al. (2018) but these are too complicated to be understood by a user who is not expert in computer science, and more particularly in machine learning. We strongly believe that the introduction of explanatory labels will help the operators understand the decisions by giving, for example, the list of speech characteristics shared by two recordings/actors. Such labels could also be used to express queries in voice search systems. Unfortunately, voice casting is an underdefined task Bonastre (2019). The operators themselves cannot formally define their decision criteria. There is also no standard taxonomy in cinema to describe voices, making it impossible to create an annotation protocol. The question is then: how can we extract features from voices when we have no a priori knowledge of them? Different representation learning approaches exist in the literature, but none of them corresponds to our expectations. 1) Feature extraction surveyed in Bengio et al. (2014) consists in representing an observation by a vector of reals but not in the form of binary vectors that we call characteristics in this paper. 2) Supervised methods like Hu et al. (2017) require labels that we do not have access to in order to extract characteristics. 3) Unsupervised methods like Peri et al. (2020) propose generic representations from which we could not extract characteristics dedicated to the domain of acted voice and character identity.

Hence, to address the lack of method to extract characteristics without ground truth, this article proposes a totally new approach entitled Label Refining. It stats with training a representation extractor using non-expert initial labels. A clustering algorithm is then employed with the purpose to emphasizing characteristics shared by groups of observations which we term refined labels. A last step consists in extracting a new representation from refined labels and comparing it with the initial representation by evaluating it on a downstream task. Unlike generic representations that use no label, this method has the advantage of guiding the extracted representations by using the non-expert initial labels.

To validate this method, we apply it to dubbing and more specifically to the extraction of characteristics from actors' voices. We therefore $p$-vector features from characters played in a video game. We apply a clustering algorithm, $k$-means, in the $p$-vector space to extract refined labels. In order to evaluate the quality of the refined labels, we train a neural network to extract refined $p$-vectors. We then subject them to a voice character similarity task to compare them with the initial $p$-vectors and thus measure the loss or gain of dedicated character information. In this experiment, we test the following hypothesis: during the clustering step, using the recordings of the training corpus only highlights the initial labels, the characters. During this step, a second corpus (close to the domain) is needed to highlight the vocal features extracted by the $p$-vectors. Since distance plays an essen-

tial role in the $k$-means algorithm, we also explore the use of cosine and Mahalanobis distances to compare with Euclidean one.

Section 2 presents Label Refining. We define the term "refined label" and present the method. The experimental protocol to apply Label Refining is described on Section 3. In Section 4, we present experiments and discuss about the obtained results. For reproducibility reasons, scripts and models are available on GitHub[1]. Finally, we conclude by presenting some takeaways and possible directions for future work in Section 5.

## 2 LABEL REFINING

Before experimenting, we first introduce the Label Refining method. It consists in extracting refined labels from initial ones. In this section, we give a definition of refined labels and we detail the method of extraction and evaluation.

### 2.1 THE DEFINITION OF REFINED LABELS

In this work, we assume that labels are decomposable into various lower-level sublabels. In our case, we wish to use the knowledge provided by the played character labels associated to the recordings with the aim to infer the characteristics such that the emotion played (e.g. agressive, happy), a voice color (e.g. white, red) or a timbre (e.g. husky, enlightened).

Equation 1 formalizes the definition of what a refined label is. Let $R$ the set of all the refined labels and $a_i^{(l)} \in \{r_1, r_2...r_{|R|}\}$ the $i^{th}$ refined label associated to the initial label $l$. Let us also specify that the number $n$ of refined labels is different depending on $l$. Thus, from the knowledge given by initial labels in $L$ we wish to find the refined labels $a_i^{(l)}$ of each $l$.

$$l \begin{cases} a_1^{(l)} \\ a_2^{(l)} \\ ... \\ a_n^{(l)} \end{cases} \tag{1}$$

Let us note the functions $f : e \mapsto l$ and $g : e \mapsto r$ where $o$ is an observation (e.g. vocal recording). The equation 2 gives fictives examples. It shows the expected images of the functions $f$ and $g$ according to two initial labels $l_1$ and $l_2$. In our model of the problem, we assume that a refined label $r$ may appear in observations associated to different initial labels. Here, this is the case with $r_1$ which appears both in an observation associated to $l_1$ and an observation associated to $l_2$.

$$l_1 \begin{cases} f : o_1^{(l_1)} \mapsto l_1 \\ f : o_2^{(l_1)} \mapsto l_1 \end{cases} , l_1 \begin{cases} g : o_1^{(l_1)} \mapsto r_1 \\ g : o_2^{(l_1)} \mapsto r_2 \end{cases} , l_2 \begin{cases} f : o_1^{(l_2)} \mapsto l_2 \\ f : o_2^{(l_2)} \mapsto l_2 \end{cases} , l_2 \begin{cases} g : o_1^{(l_2)} \mapsto r_1 \\ g : o_2^{(l_2)} \mapsto r_3 \end{cases} \tag{2}$$

### 2.2 OUR PROPOSITION TO REFINE LABELS

Figure 1 presents the refined label extraction method that we propose in three steps. ① consists in training a system to extract a representation (i.e. vector, hash Roussev (2011)) of the observation $o \in O$ using the initial labels $l \in L$ (e.g. label of the character played in a recording). ② consists in using a clustering algorithm in order to associate new labels $r \in R$ to the observations $O$. ③ consists finally in training an new system to extract refined labels by using labels $r$.

In order to evaluate the relevance of labels $r$, we use two methods. The first method, taking place at step ②, consists in applying different clustering quality measures such as $v$-measure Rosenberg & Hirschberg (2007) or the purity Ajmera et al. (2002). Applied to the step ① and step ③, the second method consists in extracting a representation of the observations $O$ and evaluating it on

---

[1] To keep the identity of the authors secret for the reviewers, the GitHub URL will be added if the paper is accepted.

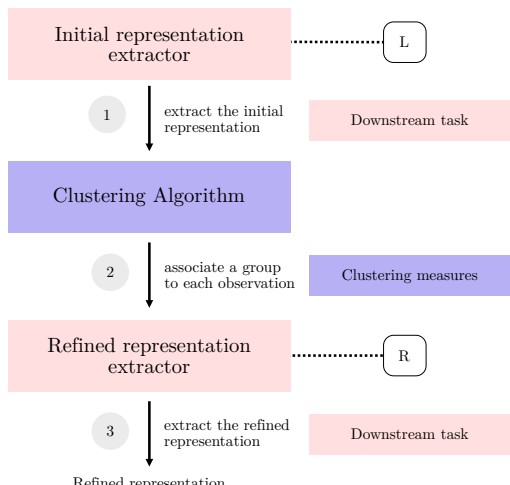

Figure 1: Refining process of $L$ initial labels into $R$ refined labels.

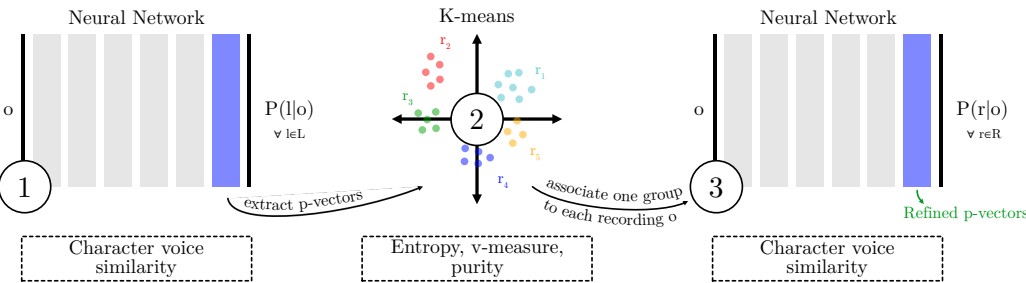

Figure 2: Refining process of character labels $L$ into $R$ refined labels applied on recordings $O$.

a downstream task. The downstream task can be diverse: speaker recognition, speech recognition or, in our case, character voice similarity. Thanks to the second method, it is possible to highlight the gain or loss of information by comparing the results obtained at steps 1 and 3. In this work, we postulate that the refined labels are pertinent if they do not lead to a loss of information when extracting refined representation.

Depending on the clustering algorithm, it is likely to have to try different values of $k$. In the following experiments, we use $k$-means and test different values of $k$. In this context, we apply the method for all the $k$ values and compare the obtained results. We postulate that Label Refining is validated if the number of refined labels $k$ differs from the number of characters and if the character information is at least preserved in the refined representation.

## 3    EXPERIMENTAL PROTOCOL

In the previous section, we introduced the generic method for extracting refined labels. We describe in this Section the protocol we follow to validate the method. We apply Label Refining on acted voice recordings in order to extract voice characteristics dedicated to the character played. A resume of the protocol is presented on the figure 2. Each step is detailed on the next subsections.

### 3.1    MAIN (MASSEFFECT) AND SECOND (SKYRIM) CORPORA

We use MassEffect data as the main corpora to train feature extractor and evaluate our system on downstream task. We also explore the use of a second dataset from the video game Skyrim during

the second step (i.e. clustering algorithm) with the aim to emphasize voice characteristics models by previous trained $p$-vectors. This subsection present both MassEffect and Skyrim datasets.

### 3.1.1 MASSEFFECT, THE DATASET USED TO TRAIN AND EVALUATE $p$-VECTORS

The main corpus is composed of voice recordings coming from the *Mass-Effect 3* role-playing game. Contrary to movies, voices of video games present some particularities explained in Gresse et al. (2019) (radio effects are present in the original voices of our corpus). They are also easier to collect since they are separated from the ambient sounds even in their final form, contrary to the dubbed film archives. Originally released in English, the game has been translated and revoiced in several other languages. In our experiments, we use the English and French versions of the audio sequences, representing about 7.5 hours of speech in each language. Segments (or recordings) are 3.5 seconds long on average. A character is then defined by a unique French-English couple of two distinct speakers. To avoid speaker identity biases, we consider only a small subset where we are certain that none of the actors play more than one character. A single audio segment corresponds to a unique speaking slot from an actor in a particular language. We then apply a filter that keeps only recordings longer than 1 second. Finally, we only keep the 16 characters for which we have the largest number of recordings, as Gresse et al. (2019) did.

In order to measure the ability of our systems to generalize, we propose a breaking down protocol. We only keep 12 characters, and their 24 corresponding speakers, for the training phase and we use the remaining 4 characters for the testing phase. We break the corpus down into training, validation and test sets. The *train* and *val* sets are composed of different recordings, but coming from the same 12 characters. To build the *train*, *val* and *test* subsets, we randomly select for each character 144, 36 and 180 recordings respectively, while balancing the number of French and English recordings. We then have a total of $1,728$ (*train*), $432$ (*val*) and $720$ (*test*) recordings. We name the three sets $S_n$.

### 3.1.2 SKYRIM, A SUBSIDIARY DATASET FOR EMPHASIZING VOICE CHARACTERISTICS

Skyrim is also a known video game. We use its recordings as a second corpus in addition to the one of MassEffect previously presented. This dataset can be viewed as a validation set. We propose to use this corpus to emphasize the characteristics of voices. Indeed, the validation set of MassEffect suffers from a lack of variability since it contains the same characters as the training set. Unfortunately, we do not have available MassEffect data with unknown characters to add to the validation set since they are used on the test. Thus, we assume that using Skyrim as a second validation set will help to bring more variability and highlight voice characteristics extracted from the $p$-vectors trained on MassEffect. This subset is composed by $18,250$ recordings of different 32 characters. As on MassEffect, each recording played in english is associate to a french one and vice-versa but we don't balance the number of recording by character.

## 3.2 EXTRACTING VOICE REPRESENTATION FROM CHARACTER LABELS

This part of the protocol corresponds to the step 1 described in the Figure 2. In order to extract the $p$-vectors, we follow scrupulously the best architecture described in Quillot et al. (2021b). Training this architecture need to follow two steps: 1) training a speaker identification system following a $x$-vector architecture on Voxceleb 2 corpora (Chung et al. (2018)) 2) replacing the output softmax of this system with $p$-vector layers, and training it to recognise the characters of the MassEffect dataset.

### 3.2.1 REFINING LABELS WITH K-MEANS ALGORITHM

This part of the protocol corresponds to the step 2 described in the Figure 2. In this work, we wish to validate our method described in 2.2. For this purpose, we focus on the $k$-means algorithm. Unlike others, this algorithm is easier to comprehend. We use the implementation of the *toolkit sklearn* with the default parameters, namely: $k - means + +$ as initializer of the barycentres in order to converge more quickly to the solution. The max number of iterations is 300 of non convergence. The algorithm is runned 10 times, then we select the clustering model with the best intertia.

| | point of view | $\lim_{k \to 1} f(C; X)$ | $\lim_{k \to |X|} f(C; X)$ |
|---|---|---|---|
| entropy | group | $+\infty$ | 0 |
| homogeneity | group | 0 | 1 |
| group purity | group | 0 | 1 |
| completeness | class | 1 | 0 |
| class purity | class | 1 | 0 |
| $v$-measure | hybrid | $\times$ | $\times$ |
| purity-$K$ | hybrid | $\times$ | $\times$ |

Table 1: Functional study of clustering measures

### 3.3 EXTRACTING REFINED VOICE REPRESENTATION BY USING CLUSTERING GROUPS

This part of the protocol corresponds to the step 3 described in the Figure 2. Once the clustering algorithm is complete, we build for each $k$-value a new neural network using the same neural architecture as used for the $p$-vector. Instead of predicting the initial labels, this new network is trained to predict the refined labels $r \in R$ using the clustering algorithm. After training, we can extract from its last layer a representation of the provided record that we name $p$-refined-vectors. The used architecture is exactly the same as for the step 1 since this neural network is also firstly pre-trained on a speaker identification task by using Voxceleb 2. Only the last layer of the $p$-vector network is changed to correspond to the number $k$ of groups (of the refined labels $r$) instead of the number of character labels $l$.

### 3.4 EVALUATION

In this subsection, we present the two ways to evelute the quality of the refining. The further one is computing clustering metrics. The second one is evaluating the representation extracted using the labels on a downstream task. We give all the details in this subsection.

#### 3.4.1 COMPUTING CLUSTERING METRICS

To evaluate a clustering model, we use the following measures: entropy, v-measure Rosenberg & Hirschberg (2007), class purity, group purity, purity-K Ajmera et al. (2002). These measures require to know for each entity in the database the group to which the algorithm has associated it as well as its original label. Since their formal definition are defined in the cited papers, we only present below their meaning and behaviour.

These measures are calculated from two different perspectives. The entropy, homogeneity and group purity measures check whether each group is associated with entities belonging to the same class. They provide a group view. The completeness and class purity measures check if each class is associated to entities belonging to the same group. They provide a class view.

To analyze the behavior of these measures, let $C$ a clustering model learnt on the observations $X$ with parameter $k$, the a priori number of clusters. Let $f$ the function which measure the quality of the model $C$ (e.g. entropy) on $X$. The table 1 shows the behaviour of the measures by computing for each of them the limit of $f$ i $k \to +\infty$ et $k \to 0$.

These measures all have their meaning and are generally misinterpreted. This is particularly the case for the entropy applied to clustering algorithms. The latter is a measure similar to that of homogeneity that has been inverted. Although it helps to compare two systems, it has a major drawback. It is bounded between 0 and $+\infty$. Interpreting the measure of a single model is therefore difficult unlike homogeneity which is bounded between 0 and 1. The latter thus gives a rate of homogeneity that entropy cannot give. Without going into details here, the group purity is a measure very similar to the homogeneity.

The measures of completeness and class purity, in addition to being very similar, are the antithesis of the previous measures. They are no longer measures of group homogeneity but measures of class homogeneity. The closer the value is to 1, the more the class in question is observed in only one group. The closer the value is to 0, the more the class in question is observed in a large number of groups at similar quantities.

Hybrid measures are weighted averages of class and group measures. Their limitations in $k \to +\infty$ and $k \to 0$ are not clear as they depend on the balance of power between homogeneity (group purity) and completeness (class purity).

Demonstrating the limits of each function is quite simple. For homogeneity and group purity, it is maximal when the elements of each group belong to the same class. When we reach $k = |X|$, each element is then assigned to a cluster and each cluster is assigned to an element. There is then only one class for each group. The homogeneity is thus maximal, the group purity too. In the opposite case, when $k = 1$, all observations belong to the same group. The group is then maximally heterogeneous and the values of homogeneity and group purity then tend towards 0. For completeness and class purity, the reasoning is similar and then becomes trivial.

### 3.4.2 EVALUATING VOICE REPRESENTATION

As described in Quillot et al. (2021b), we evaluate $p$-vectors on a character voice similarity task using a character-similarity measure based on a Siamese network. We generate *Target* trials composed of pairs of recordings belonging to the same character and *Non-target* trials made up of recordings belonging to two different characters. To avoid any bias, the number of *targets* and *non-targets* is balanced, as well as the number of pairs between two actors. So, we generate $124, 416$ (*train*), $7, 776$ (*val*) and $64, 800$ (*test*) *trials* using $S_n$ dataset (MassEffect). Skyrim is not used during this evaluation step. The threshold is set *a posteriori* at the Equal Error Rate (EER) point. System performance is then expressed in terms of accuracy on the test.

## 4 EXPERIMENTS AND RESULTS

In the experiments we present in the following sections, we decided to test different values of $k$ in order to limit the number of calculations to be performed. Several values seemed obvious to test such as 12, 24, 4 and 8 which are the number of characters and speakers respectively in the train and test. We also assume that the number of speech features we are looking for is higher than the number of characters our system has to learn to model. We therefore also tested the values 18, 32, 48 and 64 which are arbitrary values but greater than 12. On the other hand, we have also tested $k = 2$ and $k = 6$ as values less than 4 and 12 as a counter-hypothesis.

With the aim to compare results obtained on the refined $p$-vectors, we first evaluate original $p$-vectors. These $p$-vectors are trained by using $S_n$. They are evaluated by training 10 siamese neural networks and selecting the best on the validation. The best model gives us $0.69$ of accuracy on the test. This corresponds to the actual state-of-art performance presented in Quillot et al. (2021b).

### 4.1 REFINING CHARACTER LABELS BY USING MASSEFFECT

In Table 2, we plot the observed metrics *v-measure* and *purity-K* on our clustering models computed on the training dataset of $S_n$ (MassEffect). We then train 10 siamese neural networks and select the best on the validation. Its accuracy score on the validation and the test are plotted on the last two lines. For the three metrics presented in the table, the higher the value, the better the system.

On the validation set, the metrics ($v$-measure and purity-K) highlight the 12 characters contained in it and known to the system. Indeed, we naturally have higher scores for $k = 12$ with $0.91$ for the $v$-measure and $0.90$ for the purity-K.

On the test, we observe a somewhat different phenomenon. Remember that the test is composed of 4 characters. If the points – which are projections of the recordings into $p$-vector space – were perfectly discriminated by the neural classification system, they would then be organized into groups of records belonging to the same character. We would therefore have stronger measure scores for $k = 4$. In our case, we can observe this phenomenon with a loss due to the difficulty for the system to separate the records of characters that are unknown to it. Indeed, the highest values are at $k = 6$ for the $v$-measure with $0.58$ and at $k = 2$ for the purity with $0.70$. These groups are finally the closest to 4 and show a loss of quality of the representations on unknown data. However, they do not show, a priori, the existence of features that could interest us.

Table 2: Results obtained for each $k$ value by using the recordings of MassEffect for the whole label refining protocol. The $v$-measure and purity-K measures are computed on the clusters extracted from the $p$-vectors. Accuracies are computed on validation (siamese val) and test (siamese test) sets of $S_n$ by using the siamese networks trained with the refined $p$-vectors.

| | 2 | 4 | 6 | 8 | 12 | 18 | 24 | 32 | 48 | 64 |
|---|---|---|---|---|---|---|---|---|---|---|
| $v$-measure val | 0.40 | 0.62 | 0.73 | 0.80 | **0.91** | 0.86 | 0.83 | 0.80 | 0.75 | 0.74 |
| $v$-measure test | 0.54 | 0.56 | **0.58** | 0.50 | 0.48 | 0.45 | 0.44 | 0.44 | 0.41 | 0.41 |
| purity-K val | 0.41 | 0.53 | 0.64 | 0.73 | **0.90** | 0.78 | 0.68 | 0.61 | 0.52 | 0.48 |
| purity-K test | **0.70** | 0.65 | 0.64 | 0.54 | 0.50 | 0.45 | 0.43 | 0.40 | 0.32 | 0.31 |
| siamese val | 0.67 | 0.73 | 0.80 | 0.87 | **0.94** | 0.92 | 0.93 | 0.93 | 0.92 | **0.93** |
| siamese test | 0.55 | 0.65 | 0.62 | 0.66 | 0.66 | 0.69 | 0.63 | 0.66 | 0.63 | **0.70** |

Finally, we are interested in the performances obtained on the Siamese networks that we have trained. Recall that refined $p$-vectors are provided as input to these Siamese networks which are trained on a character verification task – character voice similarity. These vectors are extracted from a neural network trained to recognize the clustering group (refined label) corresponding the given recording (observation). Without surprise, we observe a best accuracy for $k = 12$. This corresponds to the number of characters. Contrary to the previous metrics, we do not observe a normal distribution in the results. Indeed, when $k$ is moving away from 12, the obtained accuracy is not necessarily lower. For instance, at $k = 64$ the obtained accuracy is not significantly lower than the one obtained at $k = 12$. Even more interesting, the accuracy obtained on the test subset is about $0.70$ that is higher than the state-of-the-art system, $0.69$.

## 4.2 Refining character labels by using Skyrim as a subsidiary corpus

In a second experiment, we wish to verify if it is possible to use a subsidiary corpus to extract voice characteristics. For this purpose, we propose this time to apply the clustering algorithm on the $p$-vectors representations extracted from the Skyrim corpus for step 2. The $p$-vectors have learned to highlight the played character but it is impossible to bring out the vocal characteristics it extracts to perform this classification. We hope that the use of a secondary corpus such as Skyrim will create enough variability to bring out vocal characteristics from the initial labels. For step 1, the $p$-vector extractor is exactly the same as the previous experiment. It is trained on MassEffect with $S_n$ and the Skyrim data is completely unknown to it. For step 2, we extract $p$-vectors using Skyrim data and perform on them the $k$-means algorithm. For step 3, we finally associate a cluster to each recording of MassEffect and train refined $p$-vectors as realized in the previous experiment.

Table 3 presents the obtained results. We note $\times$ as accuracy result when labels are missing when we associate them to MassEffect data with the aim to train the step 3 neural network. We can observe that this time the $v$-measure and the purity-K maximum scores move away from the number of characters that we originally had – 32 for Skyrim. So we know that the semantics of these clusters differ from the semantics of the character labels. In this example, purity-K highlights $k = 2$. In the $k = 2$ case, the two groupings primarily separate men and women. Each group is therefore the gender characteristic of the voice. For the $v$-measure, the score correlates with the number $k$ of groups. The best score is therefore for $k = 64$, but it might have gone up by increasing $k$. Unlike $k = 2$, it is difficult to understand the semantics of each group and to associate words that make sense to a human operator. Note that these measures have no known bias unlike the previous experiment since here the number of characters in the Skyrim data is 32 and the $k = 2$ and $k = 64$ values are significantly off.

## 4.3 Assessing the impact of the distance selected for k-means

As we know, distance in the $k$-means algorithm plays an essential role. For this reason, we propose to realise the same experiment as presented in the previous section (4.2) but using *cosine* distance instead of Euclidian one. We also use the Mahalanobis distance $k$-means presented in Lapidot (2018) – without constraints to gain computing time.

The table 4 presents results obtained by using the cosine and the Mahalanobis distances. We can observe that the Mahalanobis distance offers superior results on $v$-measure and purity-K compared to

Table 3: Results obtained for each $k$ value by replacing MassEffect 3 with Skyrim recordings during the step 2. The v-measure and purity-K measures are computed on the clusters extracted from the $p$-vectors. Accuracies are computed on validation (siamese val) and test (siamese test) sets of $S_n$ by using the siamese networks trained with the refined $p$-vectors.

|  | 2 | 4 | 6 | 8 | 12 | 18 | 24 | 32 | 48 | 64 |
|---|---|---|---|---|---|---|---|---|---|---|
| v-measure | 0.23 | 0.27 | 0.30 | 0.33 | 0.36 | 0.37 | 0.38 | 0.40 | 0.41 | **0.42** |
| purity | **0.26** | 0.23 | 0.23 | 0.24 | 0.24 | 0.24 | 0.23 | 0.23 | 0.22 | 0.22 |
| siamese val | 0.67 | 0.72 | 0.78 | 0.84 | 0.87 | 0.90 | 0.90 | 0.90 | **0.91** | × |
| siamese test | 0.50 | **0.72** | 0.65 | 0.63 | 0.67 | 0.62 | 0.66 | 0.66 | 0.62 | × |

Table 4: Results obtained for each $k$ value by replacing MassEffect 3 with Skyrim recordings during the step 2. Cosine and Mahalanobis distances are used instead of Euclidean. The $v$-measure and purity-K measures are computed on the clusters extracted from the $p$-vectors. Accuracies are computed on validation (siamese val) and test (siamese test) sets of $S_n$ by using the siamese networks trained with the refined $p$-vectors.

|  | 2 | 4 | 6 | 8 | 12 | 18 | 24 | 32 | 48 | 64 |
|---|---|---|---|---|---|---|---|---|---|---|
| cosine distance | | | | | | | | | | |
| v-measure | 0.23 | 0.27 | 0.30 | 0.32 | 0.35 | 0.37 | 0.38 | 0.39 | 0.41 | **0.41** |
| purity-K | **0.25** | 0.24 | 0.23 | 0.24 | 0.24 | 0.24 | 0.23 | 0.23 | 0.23 | 0.21 |
| siamese val | 0.70 | 0.71 | 0.78 | 0.84 | 0.88 | **0.91** | 0.90 | 0.90 | × | × |
| siamese test | 0.56 | **0.70** | 0.63 | 0.63 | 0.66 | 0.67 | 0.69 | 0.66 | × | × |
| mahalanobis distance | | | | | | | | | | |
| v-measure | 0.22 | 0.30 | 0.31 | 0.36 | 0.40 | 0.43 | 0.47 | 0.52 | 0.54 | **0.55** |
| purity-K | 0.26 | 0.25 | 0.24 | 0.26 | 0.29 | 0.30 | 0.32 | **0.36** | 0.36 | 0.35 |
| siamese val | 0.63 | 0.69 | 0.74 | 0.77 | 0.77 | 0.78 | 0.80 | **0.80** | × | × |
| siamese test | 0.59 | 0.59 | 0.56 | 0.61 | 0.54 | 0.45 | 0.55 | **0.64** | × | × |

the Euclidean distance and the cosine distance. Moreover, while the cosine distance tends to draw the same conclusions as the Euclidean distance through these measurements, the Mahalnobis distance differs in the number of clusters it highlights with the purity-K measure. Indeed, the two other distances put forward $k = 2$ while this one puts forward $k = 32$. The training of Siamese networks seems to agree with the choice $k = 32$ for Mahalnobis. However, this accuracy is significantly lower than the one obtained with cosine distance, $0.64 < 0.67$. Mahalanobis does not seem to fit the data so as to conserve character information.

## 5 CONCLUSION

This article introduces Label Refining, a semi-supervised method that consists in extracting refined labels from initial ones. It meets the needs of characteristics extraction, especially vocal, without a priori knowledge. To validate this method, we applied it to dubbing and more specifically to the extraction of characteristics from actors' voices. A first experiment showed that using the same data during all the steps of Label Refining will only highlight the initial labels. A second experiment showed that using a subsidiary corpus during the clustering step permits to highlight refined labels. It might also be noted that, in one case, our method brought to light gender characteristics. In the other case, giving labels intelligible meaning required further analysis. A last experiment showed that the Euclidean distance seems to fit the data better than cosine and Mahalanobis distances since it preserves the character information better.

With these experiments, we can conclude that Label Refining permits to extract refined labels that deviate from the original ones when using a subsidiary corpus. These labels conserve or bring the wanted information, namely character information in our case. While it is not straightforward to give meaning to these characteristics without a perceptive study, we propose in future works to build explainability methods based on Label Refining, especially for voice similarity systems.

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
