# OpenReview forum: "Label Refining: a semi-supervised method to extract voice characteristics without ground truth"
_ICLR.cc/2022/Conference — ICLR 2022 Submitted_

### Official Review · Reviewer_LKP9 · 2021-11-02

**Correctness:** 3
**Technical Novelty And Significance:** 1
**Empirical Novelty And Significance:** 1
**Recommendation:** 3
**Confidence:** 3

**Main Review:**

Pseudo labeling or iterative training with embedding feature clustering might be of related works.

Also, the author argues that the explainability of the voice recommendation system is important, however, the proposed model cannot give the controllability to the user.

Backbone network is not well explained.

Loss function of the siamese network used in the paper is not specified.

Training details are missing.

**Summary Of The Paper:**

The paper tackles a voice similarity system task for voice casting problem. The authors trained voice embedding network using voice character label and cluster the embedding features and used these clusters to train final voice embedding network. The introduction is well-written, however, the proposed method is known or marginal improvements from the existing technical skill in machine learning community (pseudo labeling with embedding feature clustering).

**Summary Of The Review:**

The problem definition seems interesting and the application the authors tackling (voice search system for voice casting) sounds interesting (introduction is well-written), however, the proposed method seems not enough novel and some details of the experimental setup is missing.

---

### Official Review · Reviewer_eoTk · 2021-11-03

**Correctness:** 2
**Technical Novelty And Significance:** 2
**Empirical Novelty And Significance:** 2
**Recommendation:** 3
**Confidence:** 4

**Main Review:**

Strengths
-------------
The paper has an interesting angle on combining data-driven and explainable (accessible by humans) clustering. The application is interesting and novel.
The authors indicate they will release source code.

Weaknesses
----------------
The paper does not contain a sufficiently detailed explanation of the underlying algorithm, and how the parameters are learned. Most of the discussion around related work is on determining voice characteristics and measuring the quality of data-driven voice representations, there is little discussion of machine learning algorithms. For ICLR audiences, it would be beneficial to expand the discussion of related work in ML.
It would be useful to provide examples of the labels used, and how they get refined during the process. As written, the description of the paper is very abstract, but I have little intuition as to how the procedure works.
The paper compares the results to a previous paper (Quillot, 2021b), and states that the proposed method achieves SoTA performance. It would be useful to include an ablation study and discuss in more detail how the approaches taken in these two papers differ, and how the design choices influence performance. The paper presents the main experiments as a series of pragmatic choices, but I am not sure if the proposed architecture has already been applied to other problems, and/ or if other choices would yield better results?
I am not sure if the data is available, so that the released source code could actually be run and the results could be reproduced?

**Summary Of The Paper:**

The paper presents a "label-refining" technique, which helps users pick voices to provide a better user experience in dubbed video games. The idea is that a voice talent's voice in the target language should match the character's voice characteristics in the source language. The method seems to work by attaching labels to data-driven clusters, and refining these using a second corpus, on which the labels' value for discrimination is measured.

**Summary Of The Review:**

The paper presents an interesting and well-justified task, but in the present form it is most suitable to a domain specific (applied) venue, rather than a more general conference such as ICLR. Because of missing comparisons and ablations, there is little information in the paper that I am confident can be generalized to other tasks and/ or problems, so I am not sure the paper will be valuable to someone not working in that field.

---

### Official Review · Reviewer_dJJ2 · 2021-11-03

**Correctness:** 2
**Technical Novelty And Significance:** 2
**Empirical Novelty And Significance:** 2
**Recommendation:** 3
**Confidence:** 3

**Main Review:**

### Strengths:

S1) The approach is simple, and seems to improve accuracy a bit over state of the art.


### Weaknesses:

W1) Overall I found the paper rather difficult to read. Besides some typos and opportunities for English usage improvement (see my comments below), I think the description of the type of information that the p-vectors carry is rather vague. In the introduction, the method is motivated by "[Voice Casting operators] do not trust the computed recommendations and need a more complete interface to interact with the system." But I don't really see how the provided approach solves this problem, because the proposed approach is still just extracting vectors and are fed to a machine learning classifier, and are thus not really any more interpretable.

W2) I'm still a little unclear what the "non-expert initial labels" are here. Are these some suggested pairs of speakers that are judged to be compatible in terms of sounding similar? This needs to be made more clear. Also, if it's indeed the "unique French-English couple" that provides these initial labels, isn't this quite subjective, since it's a decision the voice casting directors made, and doesn't necessarily imply that the voices are actually similar?

W3) I'm not quite sure what conclusion is being drawn from the second experiment where Skyrim data is used in the clustering. It seems to be that the preference towards K=2 suggests a separation in terms of gender?

W4) I think this paper would be improved by actually measuring the outcome of the task of interest by doing an evaluation with human subjects. E.g. if humans are presented with a suggested dubbing actor for an original utterance, does this method actually improve over prior approaches in terms of presenting similar voices that are suitable for dubbing?


### Minor comments

M1) "to describe comedian voices": comedian? Do you mean actor?

M2) "Vocal Casting": don't capitalize.

M3) "It stats" -> "It starts"

M4) "We therefore p-vector features": missing words?

M5) "a voice color (e.g. white, red)": this seems a little insensitive, if referring to race

M6) "fictives examples": maybe "illustrative examples"?

M7) "f : e → l and g : e → r": should this be "f : o → l and g: o → r"?

M8) "A resume of the protocol" -> "A summary of the protocol"

M9) "We propose a breaking down protocol": probably a better word choice than "breaking down"

M10) "Skyrim is also a known video game": known to whom? Do you mean to say a popular video game?

M11) Capitalize "English" and "French" everywhere.

**Summary Of The Paper:**

This paper addresses the task of finding similar-sound voices with application to voice-dubbing (e.g. finding an actor to record dialog in English, translated from original French dialog, such that the English speaker sounds similar to the original French speaker). The paper proposes a method called "label refining". This method is based on p-vectors, a prior representation found to model similarity between characters. The method uses 1) non-expert "initial labels" to train a p-vector system 2) use k-means to cluster the resulting p-vectors and 3) use the groups learned by k-means to re-group the labels (the "refined labels"). To evaluate the method, English-French pairs of voice data from the video games Mass-Effect 3 and Skyrim are used. Performance is measured using clustering measures v-measure and purity-K, as well as accuracy on the test set (which it seems the ground-truth labels are mapping to the correct dubbing speaker). The method achieves a higher accuracy of 0.70 over a state-of-the-art system that achieves 0.69. A second experiment is performed using Skyrim data as a subsidiary corpus used to cluster, with the goal of "bringing out vocal characteristics" from the initial labels, which shifts the optimal K towards 2, which the paper claims is because the new representations start to model gender.

**Summary Of The Review:**

Overall, I recommend rejection. I didn't find the paper very clear, and the conclusions are vague and not well-supported by the experiments. The novelty seems relatively minor, since this is using the p-vectors defined before, and just adding a clustering step to their training.

---

### Official Review · Reviewer_AFut · 2021-11-06

**Correctness:** 3
**Technical Novelty And Significance:** 3
**Empirical Novelty And Significance:** 2
**Recommendation:** 5
**Confidence:** 3

**Main Review:**

This Labeling Refining described in this paper could be generally applicable in different areas. Overall this paper clearly presents the proposed approach.

I find some experimental results require further clarification. E.g.
1. For results in Table 2-4, is it possible to add standard deviation to show if improvements are significant?
2. In the last paragraph on page 7, it's said best results are not for k=4 (test set has 4 characters though). Then how to validate the trained model is robust to unseen cases?
3. I'm not very clear with what patterns could be identified for distance vs. k values vs. measures for results shown in Table 4?
4. English written could be further improved, e.g. "We therefore p-vector features", "In Table 2, we plot the observed metrics" etc. Also in the first line below equation (1), should it be "o" instead of "e"?

**Summary Of The Paper:**

In this paper authors proposed a semi-supervised learning method named Label Refining, which leveraged a clustering algorithm for computing refined labels from initial ones, for training representation extractors. The proposed approach was applied to improve characteristics extraction from actors' voices.

**Summary Of The Review:**

Overall, I think this paper needs to further improve its experimental section, with more explanation for the results.

---

### Official Review · Reviewer_acRU · 2021-11-08

**Correctness:** 2
**Technical Novelty And Significance:** 1
**Empirical Novelty And Significance:** 1
**Recommendation:** 3
**Confidence:** 3

**Main Review:**

1. The paper proposes to start with initial labels and then use k-means clustering to refine labels. Overall, I do not find the approach to be novel. Its also not clear what it is really trying to achieve by this one step of refinement.

2. The paper talks about voice similarity, dubbing/voice-casting etc. but this label refinement is never really used for these tasks in the experiments. The paper merely studies the clustering through different measures and its not clear to me what the end goal here is ? The analysis of clustering offers very limited insights. In the case of the MassEffect dataset, the clustering ends up highlighting initial labels  (expected) but when another dataset is used it ends up highlighting refined labels. These seem kind of obvious conclusions ??
The paper does not offer any interesting insights on the methods as well as on the empirical end.

3. Is it possible to do this refinement in an iterative manner ? You started with some initial labels and try to iteratively refine the labels.

4. There are some recent self-supervised speech representation learning approaches such as Hubert [R1]. Authors should take a look at this paper and see if they might benefit from it as it appears to be related to it.

5. Is it possible to do experiments on how this approach can improve voice casting ?

6. There are quite a few typos and writing errors. Please proof-read the paper. Overall, the paper is quite difficult to read.

R1: HuBERT: Self-Supervised Speech Representation Learning by Masked Prediction of Hidden Units

**Summary Of The Paper:**

The paper proposes a label refinement approach. Starting from an initial set of labels, k-means clustering is used to refine the labels. The approach is described in the context of dubbing/voice casting. The method tries to obtain voice characteristics which can be further used for dubbing/voice-casting. Experiments are shown on video game datasets MassEffect and Skyrim and they primarily investigate how different parameters of k-means (# of clusters) and distance measure affect the label refinement.

**Summary Of The Review:**

Overall, I do not find the paper interesting. The approach is not particularly novel and it is not clear how it benefits the end goal. The experiments are very shallow and do not provide very interesting insights. The paper lacks strength on both methodology as well as experiments.

---

### Decision · Program_Chairs · 2022-01-20

**Decision:**

Reject

**Comment:**

This paper investigates a semi-supervised label refining approach to searching for similar voices for voice-dubbing.  The apporach is based on generating refined labels using a clustering algorithm on the initial labels.  Therefore, better voice characteristics can be extracted and used to select a new voice in the target language that closely matches the voice characteristics of the source language.  Experiments are carried out on MassEffect as the main dataset and Skyrim as the second dataset and results show that the proposed approach slightly outperforms state of the art.  While the topic under investigation is interesting and has its value to the applications such as voice casting,  there are strong concerns raised by the reviewers.  Reviewers find the paper difficult to follow.  Some important pieces of information are either missing or only vaguely explained (e.g. non-expert initial labels,  clear interpretation of p-vectors, etc.), which greatly hinders a deep understanding of the work.  Some technical details such as network architecture and its training should be elaborated.  This paper needs some good improvement in order to get accepted.  No rebuttal is provided by the authors so all these concerns still stand.